# Effects of Anthraquinones on Immune Responses and Inflammatory Diseases

**DOI:** 10.3390/molecules27123831

**Published:** 2022-06-14

**Authors:** Dandan Xin, Huhu Li, Shiyue Zhou, Hao Zhong, Weiling Pu

**Affiliations:** 1State Key Laboratory of Component-Based Chinese Medicine, Tianjin University of Traditional Chinese Medicine, Tianjin 301617, China; dandanxin172396@163.com (D.X.); zhousy0709@163.com (S.Z.); zhzyscar@163.com (H.Z.); 2Institute of Traditional Chinese Medicine, Tianjin University of Traditional Chinese Medicine, Tianjin 301617, China; 3School of Integrative Medicine, Tianjin University of Traditional Chinese Medicine, Tianjin 301617, China; woshihuhu163@163.com

**Keywords:** anthraquinones, inflammation, immune system, inflammatory diseases

## Abstract

The anthraquinones (AQs) and derivatives are widely distributed in nature, including plants, fungi, and insects, with effects of anti-inflammation and anti-oxidation, antibacterial and antiviral, anti-osteoporosis, anti-tumor, etc. Inflammation, including acute and chronic, is a comprehensive response to foreign pathogens under a variety of physiological and pathological processes. AQs could attenuate symptoms and tissue damages through anti-inflammatory or immuno-modulatory effects. The review aims to provide a scientific summary of AQs on immune responses under different pathological conditions, such as digestive diseases, respiratory diseases, central nervous system diseases, etc. It is hoped that the present paper will provide ideas for future studies of the immuno-regulatory effect of AQs and the therapeutic potential for drug development and clinical use of AQs and derivatives.

## 1. Introduction

The anthraquinones (AQs) and derivatives, aromatic quinones containing anthracenedione or 9,10-ioxoanthracenecore, are widely distributed in nature, including plants, fungi, and insects, with effects of anti-inflammation and anti-oxidation, antibacterial and antiviral, anti-osteoporosis, anti-tumor, etc. [1]. For decades, some AQs have been developed for clinical use, including anthracyclines for cancer chemotherapy and diacerein for arthritis treatment [2]. The herbs rich in AQs include *Rheum*, *Aloe*, *and Senna* species. Rhubarb (known as Da Huang), recorded in Chinese Pharmacopoeia, is the dried roots and rhizomes of *Rheum palmatum* L., *Rheum tanguticum* Maxim. ex Balf., *Rheum undulatum*, or *Rheum officinale* Bail [3]. Rhubarb is first documented in Shen Nong’s Herbal Classic and is one of the most commonly used herbs to eliminate heat, cool blood, disperse blood stasis, dredge collateral antidotal, and clinically used to treat constipation, diabetic nephropathy, chronic renal failure, acute pancreatitis and other diseases [4]. In the past decades, researchers confirmed that the main compounds of rhubarb include AQs, stilbenes, tannins, polysaccharides, etc., and they exert numerous pharmacological activities including immuno-regulating, hepato-protective, anti-cancer, anti-microbial, anti-fungal, etc. [5]. AQs are characteristic components and the main purgative compounds of rhubarb, whose content ranges from 3% to 5%. More than 30 AQs have been isolated from rhubarb, including free type and combination type. Free AQs mainly contain rhein, emodin, aloe-emodin, chrysophanol, physcion, chrysaron, etc. Combination AQs are the glycosides combined by free AQs and glycosyl [6]. The structures of the AQs mentioned in this paper are shown in Figure 1. For medical applications, AQs derivatives have been used as laxatives, antimicrobial and anti-inflammatory agents in treating constipation, arthritis, multiple sclerosis, and cancer [1].

Inflammation, including acute and chronic, is a comprehensive responses to foreign pathogens under variety of physiological and pathological processes. Inflammation is a defense response of the body to stimuli, manifested as redness, swelling, heat, pain and dysfunction [7]. Usually, inflammation is beneficial as the body’s automatic defense response, but sometimes, an excessive inflammatory response will attack the body’s own tissues, which leads to multiple dysfunctions of the host, including metabolic syndrome, organ dysfunction, etc. [8]. Inflammation plays a role for the progression of various chronic diseases/disorders, including obesity and diabetes, cancer, cardiovascular diseases, arthritis, liver injury, inflammatory bowel disease, etc. Several AQs showed obvious anti-inflammatory and immuno-modulatory effects under different pathological conditions, which attenuate tissue damages or disease symptoms [9]. We collected relevant research from the major scientific databases (Pubmed, Science Direct, Medline, CNKI, etc.), and summarized the immuno-modulatory effects and possible mechanisms of AQs according to disease type, such as digestive system, respiratory system, central nervous system, etc. The paper is meant to provide convenience for the development and utilization of AQs.

## 2. AQs Effects

### 2.1. Immune System

The immune system is the most effective weapon to prevent pathogens from invading, recognizing and eliminating exogenous pathogens and other factors that cause internal environ mental fluctuations [10]. The immune system is divided into adaptive and innate immune systems, although in fact there are many interactions between the two [11,12,13]. Macrophages are cellular components of the immune system and participate in both innate and adaptive immunity, which are responsible for patrolling and removing pathogens, apoptotic cells, debris, etc. [14,15]. Macrophages have high plasticity, and their polarization depends on the physical location and external signals received from the microenvironment [16,17,18]. In damaged tissues, macrophages are first polarized to pro-inflammatory M1 (pro-inflammatory) phenotype to fight pathogens. Subsequently, macrophages are polarized to form a M2 (anti-inflammatory) phenotype and repair damaged tissue. Numerous studies have shown that the polarization direction depends on the type and amount of cytokine, time of exposure, and the competition for cytokines [19,20,21,22]. Many herbal products, including Rhubarb and its active compounds, have been proved to regulate macrophage inflammatory response, inflammatory mediators’ production or macrophage phenotypic switch [23].

There were many studies on the anti-inflammatory effects of AQs, especially on macrophages, including RAW 264.7 and peritoneal macrophage. In most studies, AQs have been shown to have anti-inflammatory effects in various in vitro models of macrophages. In LPS-treated murine macrophage cells (RAW264.7), various pro-inflammatory factors and chemokines are highly expressed, triggered by the activation of multiple pro-inflammatory signaling pathways. AQs showed good effects in inhibiting the production of pro-inflammatory cytokines, over-activation of signaling pathways or intracellular signal transduction. Nuclear factor kappa B (NF-κB) is a key transcription factor of immune development, immune responses and inflammation. The NF-κB responds to a number of stimuli and activates with the phosphorylation of IκB and subsequent translocation of NF-κB subunits [24]. Studies have shown that during the inflammatory response of macrophages, some AQs exert anti-inflammatory effects by regulating the NF-κB pathway and downstream signals. Hu et al. reported that aloe-emodin markedly suppressed the production of NO, IL-6, and IL-1β in LPS-stimulated RAW264.7 cells through inhibiting IκBα degradation [25]. Crosstalk between NF-κB and other signaling pathways is an important factor in inflammatory response aggravation or resolution. PPARγ, a fatty acid-activated transcription factors, could negatively regulated the activation of NF-κB by several mechanisms [26,27]. PPARγ could combine with NF-κB p65/p50 dimer or inhibit IκBα degradation to suppress the activation of NF-κB and nuclear translocation [28]. For instance, Zhu et al. reported that emodin could potently suppress LPS-induced the activation of NF-κB in PPARγ (peroxisome proliferator-activated receptor gamma)-dependent manner, and thus decreasing the production of ICAM-1, MCP-1 and TNFα [29]. Similar effects on PPARγ/NF-κB were also found of rhein and chrysophanol in RAW264.7 cells (in vitro) [30,31,32] and in radiation-induced acute enteritis (in vivo) [33].

Autophagy, also known as microautophagy, is a conserved cytoprotective process to prevent cell damage and promote survival under cellular stress [34]. The connection between autophagy and inflammation has been uncovered. For example, autophagy interacts with NF-κB pathway via regulating the degradation of IκB kinase [35,36]. Some AQs exert anti-inflammatory effects by affecting autophagy. Emodin decreased the expressions of NF-κB, p62 and p-mTOR and increased IκBα expression, autophage-associated protein LC3B II/I ratio in LPS-treated RAW264.7, which showed that emodin could possibly inhibit inflammation through activating autophagy [37].

Inflammasomes are large multi-protein complexes that respond to exogenous pathogens or endogenous danger signals, which are critical to promote caspase-1-dependent maturation of IL-1β and IL-18, as well as pyroptotic cell death [38]. Many studies have shown that the NRL3 inflammasome is one of the anti-inflammatory targets of AQs. Researchers found that in LPS plus ATP-induced RAW264.7 macrophages, rhein significantly reduced phosphorylation levels of NF-κB p65 and inducible nitric oxide synthase, as well as NALP3 and cleaved IL-1β expression [39]. ATP accumulation at sites of tissue injury is an important characteristic of inflammation, and induces the disturbance of intracellular second messengers (i.e., Ca^2+^ fluxes, cAMP) in macrophages. P2X7 receptor (P2X7R), a kind of ligand (ATP)-gated ionotropic P2X receptors expressed in all cells of innate and adaptive immunity, mediates the subsequent effects of extracellular ATP [40]. Researchers have confirmed that P2X7 receptor mediated NLRP3 inflammasome activation, cytokine production, macrophage differentiation, transcription factor activation, etc. [41]. In LPS or ATP-stimulated macrophages, AQs have an effect on P2X7 receptor-mediated signaling. In HEK293 cells (expressing rat P2X7 receptor) and rat peritoneal macrophages, Hu et al. found that rhein inhibited the P2X7 receptor activation, which was characterized by the blocking of ATP-induced cytosolic Ca^2+^ elevation and pore formation of the plasma membrane. Thus, the following ROS production, phagocytosis, IL-1β release were all reduced in macrophages [42]. Emodin showed a similar effect in ATP-treated macrophages [43].

The above studies indicate that AQs inhibit excessive inflammatory responses of macrophages through multiple pathways, including NF-κB, PPARγ and inflammasomes (Table 1). Interestingly, a growing number of studies suggest that it is not simply anti-inflammatory effects of anthraquinone, but rather a bidirectional, or homeostatic, effect for macrophage. Gao et al. accidentally found the anti- and pro-inflammatory activities of rhein in LPS-activated macrophages. Rhein inhibited NF-κB activation and sequentially production of NO and IL-6. However, in the meantime, rhein enhanced the activity of caspase-1 by inhibiting IKKβ, and increasing the IL-1β and high-mobility-group box 1 production, which suggests a possible molecular mechanism for the bidirectional regulatory effects of rhein and unanticipated side effects [44]. In addition, the bidirectional regulation of anthraquinone on macrophages is closely related to the microenvironment. In primary mouse macrophages, LPS/IFNγ induced M1-like polarization and IL-4 induced M2-like polarization. Emodin inhibited M1 or M2-like polarization by inhibiting NF-κB/IRF5/STAT1 or IRF4/STAT6 signaling, respectively, which indicated that emodin is uniquely able to suppress the excessive response of macrophages to both M1 and M2 stimuli to restore macrophage homeostasis in various pathologies [45]. Emodin 8-O-glucoside (E8G, the glycosylated derivative of emodin) showed significant immune-enhancing effects of macrophage, including inducing the secretion of TNFα, IL-6, NO and enhancing phagocytosis of apoptotic T cells. These effects were partly through upregulation of the TLR2/MAPK/NF-κB signaling [46]. These results may partly explain the different effects of Rhubarb in different physiological or pathological conditions, that is, therapeutic effects or toxic side effects.

The studies of AQs on other immune cells were relatively poor. In vitro, Hwang et al. reported that physcion induced the maturation of dendritic cells (DCs) to antigen-presenting cells (APCs) through increasing the expression of surface molecules (eg, CD40, CD80, CD86, and MHC II), and thus promoting the differentiation of Th1 cells without affecting Th2 cells [47]. In mice with autoimmune thyroiditis, emodin markedly decreased the amount of CD^3+^ CD^4+^ T cells in peripheral blood monocytes and splenic lymphocytes [48]. In human T cells isolated from the peripheral venous blood of healthy donors, emodin could induce cell apoptosis by ROS-mediated endoplasmic reticulum stress and mitochondrial dysfunction, which indicated the immunosuppressive actions of emodin [49].

### 2.2. Digestive Diseases

#### 2.2.1. IBDs

The inflammatory bowel diseases (IBDs), including ulcerative colitis and Crohn disease, are chronic relapsing inflammatory disorders of the gastrointestinal tract [50]. The intestinal epithelium is a physical barrier preventing the entry of microorganisms or harmful components into the blood circulation. Clinically, intestinal epithelial dysfunction and permeability changes are the main features of IBD [51]. Changes in epithelial barrier function lead to intestinal inflammation in UC patients [52]. Infiltration of inflammatory CD4^+^ T cell, known as T helper (Th) cells, is a key characteristic of chronic intestinal inflammation [53,54]. Enhanced Th1, 2 and Th17 responses and decreased Treg and Tr1 (Type 1 Regulatory T) are common features of T cell responses in IBD. Imbalances in Th1/Th2 and Th17/Treg are considered to be an important cause of IBD [55,56,57]. In animal colitis models induced by different stimulants, AQs showed protective effects on intestinal mucosal barrier, maintenance of T cell balance and inhibition of various pro-inflammatory factors. In animal colitis models induced by different stimulus, AQs showed protective effects on intestinal mucosal barrier, maintenance of T cell balance and inhibition of various pro-inflammatory factors. In LPS (intraperitoneal injection) induced intestinal barrier injury of rats, rhein pretreatment markedly reduced the production of proinflammatory cytokiens (TNFα, IL-1 and IL-6) and inhibited the oxidative stress via MAPK and Nrf2 signaling pathway [58]. In vitro, Rhein recovered the expression and distribution of ZO-1 and weakened MLCK expression and NF-κB activation in IEC-6 cells stimulated by TNFα, which suggested the protective effect of rhein on intestinal mucosa [59]. Imbalance of intestinal microbiota is another important factor related to the pathogenesis of UC. The disruption of the microbiome leads to a rapid growing of harmful bacteria, with the release of enterotoxin which increased intestinal permeability. At the same time, the intestinal defense function and immune regulation function decline, which may cause intestinal mucosal invasion or aggravating the disease [60,61]. In DSS (a chemical substance widely used for colitis induction) induced UC mouse model, rhein could partially reverse the gut dysbacteriosis and decrease pathogenic bacteria, which was positively related with pro-inflammatory cytokines and PI3K/Akt/mTOR pathway [62]. In another study, rhein significantly relieved DSS-induced chronic colitis, which was associated with the lower uric acid. Metabolic profiles collected by untargeted metabolomics showed that rhein alleviates colitis by modulating gut microbiota, and thus indirectly altering purine metabolism in the gut [63].

Chronic inflammation contributes to colorectal cancer development and progression. In a colitis-associated intestinal tumorigenesis model induced by AOM plus DSS, emodin decreased the incidence of premalignant lesions (adenoma) at week 3 (inflammation) through inhibition the recruitment of inflammatory cell (i.e., CD11b+ and F4/80+), expression of cytokine (i.e., TNFα, IL-1α/β, IL-6). The incidence of dysplastic lesions at week 14 (tumorigenesis) was also decreased, which confirmed that emodin could attenuate intestinal inflammation and the tumorigenesis and progression [64]. In addition, emodin showed a protective effect on jejunum. In a model of sepsis caused by cecal ligation and puncture, emodin protected the jejunum in rats with sepsis by activating JAK1/STAT3 signaling pathway and regulates apoptosis proteins (Bcl-2 and Bax) expression [65].

#### 2.2.2. Pancreatitis

Acute pancreatitis (AP), one of the common inflammatory gastrointestinal diseases caused by abnormal activation of pancreatic enzymes, can cause pancreatic complications including necrosis, abscesses, or pseudocysts [66]. Based on severity, AP can be classified as mild and self-limited to severe. About 20% of patients develop severe acute pancreatitis. Abnormally elevated serum amylase is a hallmark feature of acute pancreatitis [67]. The treatment strategy for AP is mainly symptomatic treatment, including early nutritional support, pain relief, inhibition of digestive enzyme secretion, and excessive inflammatory response, etc. [68]. Rhubarb, a species of Polygonaceae, has been used as an herbal medicine in China for many years to treat various inflammatory diseases including acute pancreatitis. Rhubarb possesses the effects of diarrhea, purging the intense heat, cooling blood, removing stasis and detoxification. It is one of the most commonly used traditional Chinese medicines in the treatment of acute abdomen (including AP) as its excellent laxative and cathartic effect [69,70]. Several clinical trials have proved the add-on effect of the crude rhubarb to trypsin inhibitor, somatostatin or early enteral nutrition in patients with acute pancreatitis [71,72,73]. Modern pharmacological studies have shown that Rhubarb and its active components can effectively alleviate pancreatic damage by reducing inflammatory mediators in patients or AP model animals, promote the recovery of intestinal function, and protect the intestinal mucosal barrier. Rhein, an active component with high distribution in pancreas tissue of AP rats, protected mitochondria in AR42J cells (pancreatic exocrine cells) via the activation of PI3K/AKT/mTOR signaling pathway and activity inhibition of AMPK [74]. Emodin showed the similar protective effects in vitro of caerulein-treated AR42J cells through p38 and JNK MAPK signaling pathway [75]. In vivo (SAP rats), emodin also attenuated mitochondria injury and neutrophils-derived ROS via inhibiting the activation of voltage-dependent anion channel 1 (VDAC1) and NLRP3 inflammasome [76]. Another report demonstrated the effect of emodin on NLRP3 inflammasome seems to also relate with P2X ligand-gated ion channel [77]. Intestinal mucosal barrier dysfunction plays a key role in the pathogenesis of severe acute pancreatitis (SAP). In a rat model of SAP by injecting 3.5% sodium taurocholate into the biliopancreatic duct, free total AQs extracted from rhubarb display the beneficial effects intestinal injury and regulation of intestinal immune function. Total AQs significantly decreased the activation of NLRP3 inflammasome, the number of Tregs and the ratio of Th1/Th2, while significantly increasing the expression of sIgA in the intestinal tissues [78]. A main component of total free AQs, emodin, inhibited the expression of miR-218a-5p and thus the activation of Notch1 and RhoA/ROCK pathways in the intestine, which may partially explained the effect of emodin on intestinal dysfunction caused by severe acute pancreatitis [79].

To sum up, AQs, including emodin, rhein, etc., can significantly improve the local pathological damage of inflammatory digestive diseases, relieve local inflammation, improve the intestinal mucosal barrier, and regulate the intestinal flora (Table 2).

### 2.3. Inflammatory Diseases of the Respiratory System

The airway and lung is frequently exposed to antigens and various pathogens, which may lead to acute and chronic inflammation in the lungs, or even systemic inflammation. Immune cells in the airways are the first line of defense against pathogens, and both innate and acquired immunity [80]. Various respiratory diseases, including acute respiratory distress syndrome, chronic obstructive pulmonary disease (COPD), asthma, and cystic fibrosis, are accompanied by varying degrees of acute or chronic pulmonary inflammation [81].

Acute lung injury (ALI) and acute respiratory distress syndrome (ARDS), respiratory diseases with high mortality rates, are evoked by a wide variety of lung injuries including severe pulmonary infection, trauma, pulmonary embolism, sepsis, drug overdose, etc. [81]. Excessive accumulation and activation of leukocytes and platelets, as well as the following activation of various signaling pathways and cytokines production, exacerbate the disease [82,83,84]. In LPS-provoked ALI and ARDS, emodin showed obvious inhibition of inflammation and maintenance of alveolar structure. In LPS-induced ALI rats or mice, emodin ameliorated pathological changes and infiltrated inflammatory cells. The levels of various inflammatory factors, including TNF-α, IL-1β, and IL-6, in bronchi alveolar lavage fluid were decreased after emodin treated [85]. The effects were associated with the inactivation of mTOR/HIF-1α/VEGF and NF-κB signaling pathways [86]. In diesel exhaust particles (DEP, air pollutant)-induced impairment of lung function of mice, emodin significantly improved pulmonary lipid peroxidation, inhibited reactive oxygen species production and the reduction of glutathione, which indicated the attenuation of oxidative stress [87].The anti-inflammatory effects in ALI of another anthraquinone, chrysophanol, appear to be related to the histone deacetylase 3 (HDAC3)-mediated HMGB1/NF-κB pathway [88]. Similar to emodin, the NF-κB pathway plays an integral role in the anti-inflammatory effects of chrysophanol.

Acute asthma is an important medical emergency, which is mainly triggered by viral respiratory infections and allergen exposure [89]. The active type 2 airway inflammations, as measured by blood eosinophils or elevated exhaled nitric oxide, is a key independent risk factor for exacerbations of asthma [90]. Asthma exacerbations associated with viral infection have been shown to be associated with neutrophilic inflammation [91]. In addition, non-type 2 inflammation is more likely to be involved in viral infection and corticosteroid resistance [92]. Besides corticosteroids and selective β2-agonists, monoclonal antibody therapy, such as IL-5 and IL-5 receptor monoclonal antibodies and Dupilumab (a monoclonal therapy against the IL-4 and IL-13), have shown therapeutic advantages in patients with severe asthma [93]. AQs exhibit significant therapeutic effects on multiple pathological processes of asthma in in vivo models. In ovalbumin (OVA)-induced bronchial asthma mouse model, emodin inhibited the infiltration of inflammatory cells and reduced the levels of IL-4, 5, 13, 17, NO and IFNγ. In particular, infiltrated macrophages and eosinophils were significantly reduced in lung after Emodin treated. This effect is associated with inhibition of activation of Notch and NF-κB signaling pathways [94,95]. Chrysophanol attenuates OVA-induced airway remodeling in mice, and expression of α-SMA. In vitro, in BEAS-2B cells (human pulmonary epithelial cells) treated with TNFα, chrysophanol exhibited anti-proliferative effects and inhibited the activation of NF-κB signaling pathway [96].

Pulmonary fibrosis is an important feature of some lung diseases, including chronic obstructive pulmonary disease (COPD) and silicosis. TGFβ, a major inflammatory mediator discharged by activated epithelial cells and macrophages, induces transition from fibroblast to myofibroblast, and EMT of alveolar epithelial cells [97,98]. Besides EMT, inflammasomes (such as NLRP3)-NF-κB signaling pathway activation and the following upregulation of IL-1β and IL-18 are involved in the pulmonary fibrosis [99]. In bleomycin-induced pulmonary fibrosis of rats, emodin significantly reduced lung structural damage, collagen deposition, massive inflammatory cell infiltration, and pro-inflammatory cytokine secretion, which was partly attributed to NF-κB inhibition and Nrf2 upregulation. In vitro, emodin suppressed the TGFβ1 induced EMT-like shifts of alveolar epithelial cells [100]. In silica-induced lung injury mice (Silicosis), emodin reduced the degree of alveolitis and fibrosis in the lungs induced by silica particles. Emodin slows the progression of pulmonary fibrosis through multiple levels of inflammatory response regulation and related protein expression, including inhibition of Smad3 and NF-κB phosphorylation, apoptosis, and EMT [101] (Table 3).

### 2.4. Bone Tissue

Various kinds of arthritis are accompanied by chronic inflammation, including rheumatoid arthritis (RA), osteoarthritis (OA), gouty arthritis (GA) and reactive arthritis [102]. There are a large number of inflammatory cells such as lymphocytes and macrophages in the joint cavity, which release inflammatory mediators and activate the immunity under pathological conditions [103]. Rheumatoid arthritis (RA), a chronic autoimmune disease with joint pathology, is characterized by persistent synovitis, hyperplasia, systemic inflammation, and autoantibodies production (particularly to rheumatoid factor and citrullinated peptide) [104]. Osteoarthritis (OA), a degenerative disease of articular cartilage, is a complex process composed of inflammatory and metabolic factors. Local (synovitis) and systemic inflammation play a key role in the pathogenesis of OA [105]. Therefore, blocking the infiltration of inflammatory cells and secretion of inflammatory mediators is the key to the treatment of arthritis. Rhein, a bioactive constituent of anthraquinone, was confirmed to have excellent therapeutic effects on arthritis. For example, in OA, rhein inhibited the production of IL-1β, IL-1ra and NO in the synovial tissue [106]. In RA, rhein inhibited ATP-induced Ca^2+^ and suppressed the production of intracellular ROS [107]. In addition, rhein regulated the formation and differentiation of osteoblasts, namely rhein reduced bone resorption and osteoclast resorption plaque by inhibiting the formation of multinucleated osteoclasts. Since Wang et al. has reviewed the effects and mechanism of rhein on arthritis, we will not repeat it here [108]. Diacerein, a prodrug of rhein, has showed great efficacy on the treatment of OA as an important IL-1β inhibitor. Clinically, diacerein has unique biological activities on inflammation of cartilage and synovial and subchondral bone remodeling. Almezgagi et al. summarized the recent points of diacerein on various inflammatory diseases, including arthritis [109]. Emodin is another well-studied AQ with significant joint protection. In collagen-induced arthritis (CIA) mice, emodin inhibited synovial inflammation and joint destruction (in vivo), and M-CSF induced osteoclast differentiation in bone marrow macrophages (in vitro), which were both related with the inhibition of the NF-κB pathway [110]. In LPS-stimulated synoviocytes (obtained from patients with RA) under hypoxia, emodin significantly inhibited IL-1β and LPS-stimulated proliferation of RA synoviocytes, as well as the production of pro-inflammatory cytokines (TNFα, IL-6 and IL-8), mediators (PGE2), matrix metalloproteinase, which were related with the reduced histone deacetylase (HDAC) activity [111]. In another research, emodin induced apoptosis and autophagy of fibroblasts obtained from patient with ankylosing spondylitis (AS) in vitro [112]. In arthritis rats induced by complete Freund’s adjuvant, aloe-emodin decreased arthritic score and various biochemical and hematological parameters, including some key oxidative stress markers, which indicated the inhibition of inflammation and arthritis [113]. The above studies have shown that AQs (especially rhein derivative Diacerein for clinical application) can inhibit joint inflammation, protect joint structure and thus alleviated disease in various types of arthritis (Table 4).

### 2.5. Metabolic Disease

The increase in high-calorie diets and the decline in physical activity have led to a rapid increase in the global incidence of metabolic diseases. Obesity, characterized by excessive accumulation of fat, is considered to be a major risk factor for various metabolic diseases, such as insulin resistance, diabetes mellitus, non-alcoholic fatty liver disease (NAFLD), atherosclerosis and hypertension [114]. Chronic inflammation of is a characteristic feature of obesity, which was largely due to the increased accumulation of proinflammatory macrophages in adipose tissue. Macrophages are the most abundant immune cell type in obese adipose tissue and liver, thus mediating insulin resistance and metabolic dysfunction of liver [115]. Obesity can lead to hepatic steatosis, known as nonalcoholic fatty liver disease (NAFLD), which can progress to nonalcoholic steatohepatitis (NASH), fibrosis, and cirrhosis, and eventually hepatocellular carcinoma. A major component of obesity-induced liver inflammation is attributed to increased monocyte infiltration and macrophage differentiation [116]. However, the mechanism of macrophage accumulation in pancreatic islets is distinctly different compared to that in liver and adipose tissue. In pancreatic islets, macrophage accumulation is mainly mediated by the proliferation of islet-resident macrophages [117]. In addition, pro-inflammatory cytokines produced by islet macrophages also suppress β-cell GSIS (glucose-stimulated insulin secretion) [118]. Emodin has been reported to exert various effects in metabolic disorder, including lipid lowering, blood glucose control, and anti-inflammation. In mice fed with a high fat diet plus LPS to mimic NAFLD, emodin ameliorated systemic inflammation, reduced inflammatory cell infiltration in the liver, and attenuated liver function impairment [119]. In normal state, resident adipose tissue macrophages mostly show an M2-like polarized phenotype, while in obesity, the M1-like polarized phenotype increased. Therefore, promoting M2-like polarization of adipose macrophages is also an important method to relieve fat accumulation. In high-fat diet (HFD)-induced obese mice, emodin significantly inhibited lipid accumulation, reduced glucose and insulin levels, ameliorated serum lipid profiles and the local and systemic inflammation. The percentage M2 macrophage was greatly increased by emodin in adipose tissue [120]. In a high-fat diet plus streptozotocin-induced T2D rat model, emodin ameliorated hyperglycemia, dyslipidemia, and glucose metabolism, which was controlled by SMAD7 regulated by miR-20b [121]. In vitro, emodin upregulated glucose metabolism, decreases lipolysis, and inhibits inflammation in C2C12 myotubes and 3T3-L1 adipocytes [122]. The AMP-activated protein kinase (AMPK) is a central regulator of multiple metabolic pathways, and the activation of AMPK is vital in increasing glucose uptake, fatty acid oxidation, and autophagy, while suppressing the synthesis of fatty acids, cholesterol, and protein. Another anthraquinone, chrysophanol, exhibited the effect of AMPK activation. In HFD rats, chrysophanol treatment decreased body weight, blood glucose and the blood level of triglyceride (TG). In vitro, chrysophanol reduced lipid accumulation and HFD-induced inflammation of primary hepatocytes, which was attributed to the activation of AMPK/Sirtuin 1 signaling pathway [123]. In the case of hyperlipidemia-induced oxidative stress and inflammation of heart, aloe-emodin significantly reduced the expression levels of proinflammatory cytokines (IL-1β, IL-6, and TNFα), as well as vascular cell adhesion molecule 1 (VCAM1) and intercellular adhesion molecule 1 (ICAM-1). In palmitic acid stimulated H9C2 (embryonic rat heart-derived cell line), AE enhanced cell viability, reduced ROS production and pro-inflammatory mediators secretion via inhibition of the TLR4/NF-κB signaling pathway [124] (Table 5). Mohammed et al. reviewed the antidiabetic potential of AQs [125], which will not be elaborated here.

### 2.6. Cerebral Vascular System and Central Nervous System

Stroke, the second leading cause of death worldwide, is mainly caused by occlusion of cerebral artery. The immune response to acute cerebral ischemia is a major factor in stroke development or prognosis [126]. Inflammation caused by stagnant blood flow, activation of intravascular white blood cells, and release of pro-inflammatory mediators may aggravate tissue damage. Although the inflammatory response begins locally in the ischemic blood vessels and brain parenchyma, the inflammatory mediators released spread throughout the organism [127]. There is growing evidence that immune responses play a dual role in stroke. In the acute phase, innate immune cells invade the brain and meninges and cause ischemic injury. At the same time, dangerous signals (such as ROS, MMPs) released by damaged brain cells into the circulatory system can trigger the activation of the immune system throughout the body, leading to severe immune suppression that even may lead to life-threatening infections. In the chronic phase, antigen presentation triggers an adaptive immune response in the brain that may be an important cause of post-stroke morbidity [128]. The upregulation of proinflammatory cytokines is a major character during stroke. IL-1 family, detected in the infarct area, is associated with brain tissue damage. IL-1β processing is regulated by a number of cytosolic factors, among which inflammasome is an important family of protein complexes [129]. Researchers found that the brain tissue damage is accompanied by NALP3 inflammasome activation [130]. In mouse transient middle cerebral artery occlusion (tMCAO) model, chrysophanol reduced neurological deficits, infarct volume, brain edema, and blood–brain barrier (BBB) permeability, which was partial related with the inhibition of NALP3 activation [131]. Another research reported that chrysophanol also protected the neuronal apoptosis or death induced by ischemia/reperfusion (I/R) cascade of MCAO mice. Chrysophanol reduced brain tissue loss, improved neurological assessment and motor function, accompanied with the reduction of TNF-α, IL-1β and NF-κB p65 in neurons [132]. These results indicated that chrysophanol attenuates brain injury after focal I/R partly attributed to its anti-inflammatory effects. Oxidative stress is another major pathophysiological pathway in the pathogenesis of stroke, which also has crosstalk with inflammation. It is confirmed that oxidative stress in neurons and microglia-induced neuroinflammation is involved in the occurrence of stroke [133]. Xian et al. found that aloe-emodin significantly improved the infarct size and behavioral score of MCAO rats, decreased the expression of TNFα, MDA, and increased SOD. In vitro, in oxygen and glucose deprivation reperfusion (OGD/R) model of human neuroblastoma cells (SH-SY5Y) and LPS stimulated BV2 cells, AE significantly protected SH-SY5Y cells from the injury of OGD/R and reduced the production of inflammatory cytokines in BV2 cells stimulated by LPS via PI3K/AKT/mTOR and NF-κB signaling pathways [134]. In H2O2-induced apoptosis and neuroinflammation of SH-SY5Y cell, emodin exerted the similar effects on oxidative stress. Emodin significantly enhanced cell viability, reduced cell apoptosis and LDH release, alleviated H2O2-induced oxidative stress and mitochondrial dysfunction by regulating the PI3K/mTOR/GSK3β signaling pathway [135].

Sepsis-associated encephalopathy (SAE) is a severe complication of sepsis and may cause cognitive dysfunction, apoptosis of neurons and neuroinflammation [136]. In SAE mice, emodin significantly ameliorated cognitive dysfunction, inhibited apoptosis and induced autophagy in hippocampal neurons via upregulation of BDNF/TrkB signaling [137]. Another study confirmed the effects of emodin on sepsis-associated acute brain injury, and demonstrated that the effect was possibly due to the activation of cholinergic anti-inflammatory pathway [138].

Multiple sclerosis is an inflammation-mediated chronic, progressive autoimmune disease that occurs in the central nervous system, resulting in demyelination of the brain and spinal cord, and nerve damage [139]. Experimental autoimmune encephalomyelitis (EAE) is an animal model for the study of MS, as its similar pathological features to MS. Rhein significantly decreased the incidence of EAE [140]. Rhein significantly reduced the level of IL-2 in the brain and spinal cord tissues of EAE mice, up regulated the expression of Foxp3. Foxp3 is a major regulator of immunosuppressive responses and a key factor in regulating Treg differentiation and function, which indicated the effect of rhein on EAE is related with immune responses of T cell [141]. To sum up, in a variety of brain injuries accompanied by acute or chronic inflammation, AQs play a protective role through different pathways or immune cells (Table 6).

### 2.7. Liver

Drug-induced liver injury (DILI), a kind of adverse drug reactions (ADRs), is one of the most challenging liver disorders in clinic [142]. DILI is usually includes intrinsic and idiosyncratic DILI. Intrinsic DILI is predictable, dose-dependent and occurs in the majority of individuals exposed to the drug; while idiosyncratic DILI is unpredictable, independent of dose, and occurs in only a small proportion of individuals [143]. Regardless of the type of DILI, it is usually accompanied by varying degrees of mitochondrial dysfunction, oxidative stress, tissue necrosis, autoimmune-like hepatitis, etc. [144]. Some research found that AQs show protective effects on acute liver injury in DILI. Methotrexate, an anti-neoplastic drug by anti-metabolism, is widely used to treat rheumatoid arthritis and cancer chemotherapy. However, as a narrow therapeutic index, Methotrexate is a highly toxic drug. Abnormal liver function is a major side effect of Methotrexate [145]. In rats with Methotrexate induced liver injury, rhein significantly reduced the level of ALT, AST and morphological damage. In vitro, rhein inhibited the apoptosis of Methotrexate -treated normal human hepatocyte (L02 cells), as well as NF-κB, TNFα and caspase-3. Rhein also activated the antioxidant pathway Nrf2-HO1, which was related with the hepatoprotective effects of rhein [146]. Acetaminophen (APAP) is the most commonly used analgesic drug for relief of pain and fever worldwide. APAP overdose leads to dose-dependent liver injury in all mammalian species, which is related with mitochondrial ROS and RNS, DNA fragmentation and cell death, and sterile inflammation, including TLRs activation and the following inflammatory signaling pathway [147]. In mice with APAP-induced liver injury, oral treatment of emodin attenuated the tissue damage through inhibiting oxidative stress via the AMPK/Hippo-Yap mediated pathway [148].

Alcoholic liver disease (ALD) is a chronic liver disease caused by excessive alcohol abuse. The initial stage is usually liver steatosis, which can develop into steatohepatitis, fibrosis and/or cirrhosis. The imbalances of lipid metabolism in liver cells mark the onset of ALD, which in turn accelerates hepatocyte damage [149]. In mice with steato-hepatitis induced by excessive alcohol, physcion attenuated lipid accumulation and inflammation in ALD mice by regulating the Bmal1 (core circadian gene) expression, a process involving AMPK/PPARα-involved fatty acid oxidation. In vitro, the effect of physcion on ethanol treated HepG2 cells was in accordance with that of in vivo [150]. Although AQs show obvious hepato-protective effects, more attention has been paid to the liver injury caused by traditional Chinese herbs containing AQs, such as Radix Polygoni Multiflori and Radix et Rhizoma Rhei. Many studies have summarized the causes of hepatotoxicity of AQs, and we will briefly elaborate the contents related to immune regulation in the discussion (Table 7).

### 2.8. Kidney

Acute kidney injury (AKI) or chronic kidney disease (CKD) caused by various factors is a serious threat to human health, especially CKD. CKD is irreversible and often evolves to end-stage renal disease. AKI or CKD accompanied by a variety of pathological changes, including changes of renal vasculature and oxygen delivery, altered tubular epithelial cell maturation and pericyte activation status, lymphocyte subtypes and functions, etc. [151]. Diabetic nephropathy (DN), a severe complication of diabetes mellitus, has a great impact on the increasing population with CKD and is a main cause of end-stage kidney disease worldwide. The progression of DN includes glomerular hyperfiltration, inflammation of glomeruli and tubulointerstitial regions and reduction of cell number and extracellular matrix (ECM) accumulation [152]. Rhubarb and its formula are frequently used for the treatment of CKD in eastern Asia countries. Emodin exerted a renoprotective effect in DN. In high glucose (HG) treated mouse mesangial SV40-MES13 cells, emodin significantly alleviated the oxidative stress, inflammation and ECM accumulation, which was partially due to the inhibition of NF-κB and PI3K/AKT signaling pathways. In addition, these pathways were modulated by the circ_0000064/miR-30c-5p/Lmp7 (large multifunctional protease 7) axis [153]. In LPS treated renal tubular epithelial NRK-52E cells (AKI), emodin also inhibited LPS-induced TLR2, NF-κB, TNFα, IL-1β and IL-6 expression in vitro, which may contribute to the immune regulation of emodin in LPS-induced acute kidney injury [154]. In the kidney, autophagy has been suggested to induce tubular atrophy, thus contribute to cell death during kidney injury. However, in some pathological conditions, autophagy plays a protective role in renal injury, which indicated the role of autophagy in CKD is controversial [155]. In vivo, rhubarb attenuated ade-induced renal tubular injury, with inactivation of autophagy and inhibition of fibrosis. In vitro, rhein inhibited autophagy of Hank’s balanced salt solution (HBSS)-stimulated NRK-52E cells in the AMPK-dependent mTOR signaling pathways. However, rhein alone induced autophagy in NRK-52E cells, which indicated the anti-autophagic effect of rhein only occurred in pathological situation. In uric acid-induced inflammatory injury of mouse kidney epithelial cell line (TCMK-1), rhein significantly decreased the levels of IL-6, IL-1β and TNFα, and cell apoptosis of TCMK-1 cells, which was regulated by lincRNA-COX2/miR-150-5p/STAT1 axis [156]. The above research indicated that emodin and rhein have different protective effects on kidney under various pathological conditions (Table 7).

**Table 7 molecules-27-03831-t007:** Effects and mechanism of AQs on liver and kidney injury.

Compound	Disease/Injury	Stimuli	Cell/Animal	Doses	Effects	Mechanism	Ref
Rhein	DILI	MTX	Wistar rats	ig. 20, 50 and 100 mg/kg	ALT, AST, morphological damage↓	NF-κB↓ Nrf2-HO1↑	[146]
Rhein	In vitro	MTX	L02	5, 10, 20 μM	TNFα, caspase-3↓		
Rhein	In vitro	Uric acid	TCMK-1	10, 20 and 40 μg/mL	TNFα, IL-1β, and IL-6↓ Apoptosis↓	miR-150-5p/STAT1	[156]
Emodin	DILI	APAP	C57B/6 mice	ig. 10, 30 mg/kg	ALT, AST↓, tissue damage↓ Antioxidant enzyme↑	AMPK-Hippo/Yap↑	[148]
Emodin	In vitro	APAP	HepG2	3–30 μM	Cell death↓, ROS↓, Mitochondrial dysfunction↓		
Emodin	In vitro	Glucose	SV40-MES13	20, 40 μm	MDA, ROS↓, SOD↑ TNFα, IL-1β, IL-6↓ FN, collogen I↓	Circ_0000064/miR-30c-5p/Lmp7	[153]
Emodin	In vitro	LPS	NRK-52E	20, 40 μm	TNFα, IL-1β, and IL-6↓	TLR2-NF-κB	[154]
Physcion	ALD	Ethanol	C57BL/6	ig. 250, 500 μg kg	Fat vacuole accumulation NLRP3 inflammasome↓	BMAL1 AMPK/PPARα	[150]
Physcion	In vitro	Ethanol	HepG2	0.125, 0.25 μm	IL-1β and Caspase-1↓ Lipid Accumulation↓		

## 3. Discussion and Conclusions

Various types of diseases are accompanied by acute or chronic inflammatory responses. AQs could alleviate pathological damage via modulating immune responses through various signaling (summarized in Figure 2). The effects of AQs on immune cells (especially macrophages) were mainly studied in vitro. AQs, including emodin and rhein, affected macrophage M1-like polarization, pro-inflammatory cytokines secretion, apoptosis and other processes through NF-κB, PPARγ, NRLP3 inflammasome, P2X7 receptor and other pathways and interactions, thus inhibiting excessive inflammation. Interestingly, some studies have shown that the effects of AQs are not simply consistent anti-inflammatory effects. For example, although rhein inhibited the production of IL-6 and NO, it promoted the secretion of pro-inflammatory factors IL-1 and HMGB1 [44]. Emodin 8-O-glucoside showed significant immune-enhancing effects of macrophage, including inducing the secretion of TNFα, IL-6, NO [46]. This differential effect seems to be related to the stimulus of the microenvironment. Emodin inhibited LPS-induced M1 or IL-4/13-induced M2-like polarization by inhibiting NF-κB/IRF5/STAT1 or IRF4/STAT6 signaling [45]. That is, AQs appear to restore macrophage homeostasis rather than simply anti-inflammatory effects, which is related to the microenvironment or pathological state/disease stage in vivo. This seems to explain the paradoxical role of AQs in the body. This hypothesis is supported by AQs-induced liver injury. The difference of liver immune microenvironment significantly affected the results of liver response to AQs. Under liver fibrosis (immune tolerance), AQs showed therapeutic effect. While in normal state (immune activated), liver is more sensitive to AQs damage, which is related to Kupffer cells (liver macrophages) [157]. This exposes a shortcoming in the current research, that is, the in vitro effects and mechanism studies are relatively isolated and cannot fully simulate the in vivo pathological state, which may lead to contradictory results between in vitro and in vivo studies.

In digestive system diseases, rhein and emodin are the most studied ones. In addition to the direct effects on the inflammatory pathways (NF-κB, MAPK, inflammasome, etc.), the protective effect of AQs on the intestinal mucosal barrier and the regulation of intestinal flora serves as the indirect mechanism. In addition, miRNAs appear to be involved in the immuno-modulatory effects of AQs through post-transcriptional regulation. In multiple types of respiratory diseases, emodin and chrysophanol have shown promising therapeutic effects in in vivo and in vitro studies. AQs alleviate lung inflammation, fibrosis and airway remodeling by enhancing NRF2-HO1-mediated antioxidant stress and inhibiting TGFβ-mediated fibrosis. In osteoarticular diseases, have shown that rhein and emodin have good effects both in vitro and in vivo through autophagy enhancement and NF-κB inhibition. In particular, the prodrug of rhein, Diacerein, has been widely used in the clinical treatment of OA. Among metabolic disorders, including obesity, T2D, and NAFLD, macrophage polarization and the AMPK pathway seems to be the major targets by which AQs regulate metabolic abnormalities. In the inflammatory injury of brain tissue caused by various diseases, AQs play a significant role in relieving nerve apoptosis and pathological changes via inhibition of NF-κB, inflammasome, PI3K/mTOR and activation of autophagy. Last but not least to mention is that the blood–brain barrier (BBB) restricts the drug development of brain-targeted drugs for AQs. In in vitro studies of AQs targeting brain tissue, most studies seem to ignore the influence of BBB on drug dosage. Due to the existence of the BBB, many compounds cannot directly reach the brain tissue. According to the TCMSP database (https://tcmsp-e.com/index.php (accessed on 1 June 2022)) we queried, the BBB values of AQs summarized in the paper are: −0.2 for chrysophanol, −0.66 for emodin, −0.99 for rhein and −1.07 for aloe-emodin. Generally, compounds with BBB values less than −0.3 are considered difficult to penetrate the BBB. Although only reference data, in vitro studies of brain-targeted AQs should take this into account.

In acute liver and kidney injury, especially DILI, AQs achieve liver and kidney protection by inhibiting NF-κB and increasing the activation of Nrf2 and AMPK signaling pathways. It is worth mentioning that AQs is also one of the factors that cause liver and kidney damage. Inappropriate use of AQs-containing hers, such as Radix Polygoni Multiflori (known as Heshouwu) and rhubarb, may induce liver injury. Metabolic enzymes alteration, hepatocyte apoptosis, bile acids homeostasis disruption, and inflammatory damage are the possible causes of liver injury [158,159,160,161,162]. As mentioned above, the effects of AQs are different in physiological and pathological states, which may be one of the focuses. In view of the multi-directional immuno-modulatory effects of AQs, it is also important to reveal the molecular mechanism of toxic and side effects as well as to clarify the effective mechanism. How to make dialectical diagnosis and treatment and adjust the dosage and course based on experimental research will be another focus of future research. For the well-studied and promising ones, including emodin and rhein, more in-depth studies are needed to reveal the molecular mechanisms of AQs. Improving bioavailability and reducing toxicity is the key for drug development while maintaining or improving efficacy.

The drug-likeness (pharmacokinetics) of AQs is one of the key factors affecting their in vivo activity and drug development. Pharmacokinetic analysis indicated that most AQs have a poor intestinal absorption, bioavailability, and short elimination half-life, while the AQ-glycosides have higher oral bioavailability [163,164]. Besides, the AQs in serum mainly presented as glucuronides/sulfates, and in tissues mainly free form of AQ. For example, the aloe-emodin and rhein are mainly found in kidney, liver, lung; emodin mainly in liver, lung; and chrysophanol in kidney and liver [165,166]. The physicochemical and ADME properties of various AQs referred from TCMSP database (https://old.tcmsp-e.com/index.php (accessed on 1 June 2022)) confirmed that the drug-likeness properties (DL) of free AQs were between 0.24 and 0.3, while the value of AQ-glycosides were mostly above 0.7. Of course, the absorption of AQs varies greatly under physiological and different pathological conditions [167,168].

The safety AQs is another important factor to be considered in their clinical application and drug development. With the widely used as medicinal herbs and foods, the toxicity of some AQs-containing herbs, such as Rheum and Polygonum, gradually emerged due to the improper use [169,170]. The toxicity of the AQs reported in the current research mainly includes hepatotoxicity, nephrotoxicity, genotoxicity, reproductive toxicity and phototoxicity [171,172]. The primary toxicity of AQs is hepatotoxicity and nephrotoxicity, which are of wide concern. Studies have shown that direct (inserted into the base pair of DNA) or indirect DNA damage (ROS medicated) and subsequent apoptosis or inflammatory responses caused by high doses of AQs are important reasons for their liver and kidney toxicity [159]. Long term usage of AQs has been linked to colonic toxicity as the accumulation of toxic metabolites, which may induce apoptosis and autophagy of colonic epithelial cells [160]. In recent years, aloe-emodin has been reported to have phototoxicity, which is mainly manifested in the damage to skin fibroblasts exposure to ultraviolet radiation. Multiple studies have shown that a photochemical mechanism involving singlet oxygen and the direct photo-oxidative damage to DNA or RNA may be responsible for the phototoxicity [162]. Thus, the drug development of AQ should focus on the direction of high efficiency and low toxicity.

To sum up, this paper reviewed the recent research of AQs on anti-inflammatory and immuno-modulatory effects on immune system, digestive diseases, respiratory diseases, metabolic disorder, etc. The main molecular mechanisms include anti-inflammatory effects centered on NF-κB inhibition, NrF2-mediated antioxidant stress, AMPK pathway involved metabolic regulation, and non-coding RNA regulated post-transcriptional regulation. It is hoped that the present paper will provide ideas for future studies of the immuno-regulatory effect of AQs and the drug development and clinical use of AQ and related herbs.

## Figures and Tables

**Figure 1 molecules-27-03831-f001:**
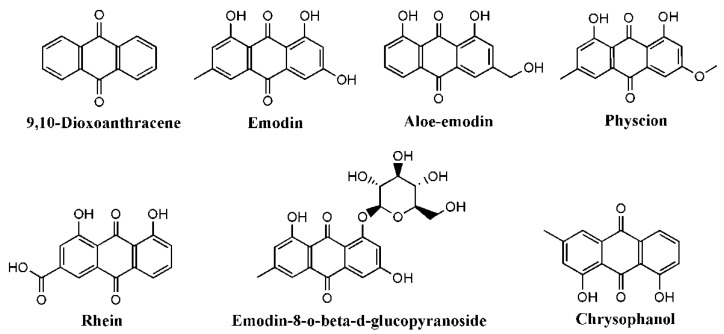
Chemical structure of AQs mentioned in this paper.

**Figure 2 molecules-27-03831-f002:**
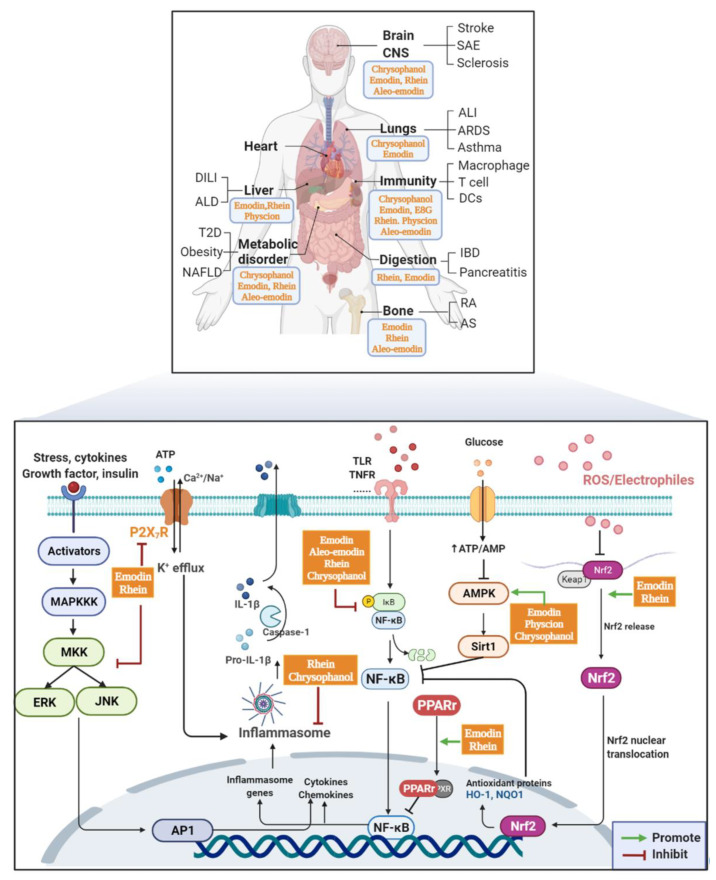
Effects and mechanism of AQs on inflammatory diseases/injury.

**Table 1 molecules-27-03831-t001:** Effects and mechanism of AQs on immune system.

Compound	Disease/Injury	Stimuli	Cell/Animal	Doses	Effects	Mechanism	Ref
Aloe-emodin	In vitro	LPS	RAW264.7	10–20 μM	NO, IL-6, and IL-1β↓	NF-κB↓	[25]
Emodin	In vitro	LPS	RAW264.7	25 μM	ICAM-1, MCP-1 and TNFα↓	NF-κB↓; PPARγ↑	[29]
Emodin	In vitro	LPS	RAW264.7	0–50 μM	TNFα IL-1β and IL-6↓	NF-κB↓; LC3B II/I (autophagy)↑	[37]
Emodin	In vitro	LPS/IL-4	Primary macrophages	0–50 μM	Phagocytosis↓; NO↓; Migration↓	NF-κB/IRF5/STAT1↓(LPS) IRF4/STAT6↓(IL-4)	[45]
Emodin	In vitro	ATP	peritoneal macrophages	0.1–10 μM	cytosolic Ca^2+^↓; phagocytosis↓ ROS, IL-1β↓	P2X7↓	[43]
Emodin	Autoimmune thyroiditis	NaI	non-obese diabetic mice	ig. 15, 75 or 150 mg/kg	serum TgAb↓ serum IFN-γ↓	CD3^+^CD4^+^ T cell↓	[48]
Emodin	In vitro	_	Primary human T cells	1–100 μM	Apoptosis↑; Ca^2+^ ROS, MDA↑; SOD↓ Caspase 3, 8, 9↑	Endoplasmic reticulum stress↑ Mitochondrial dysfunction	[49]
Emodin 8-O-glucoside	In vitro	LPS	RAW264.7 THP-1	20 μM	TNFα, IL-6, NO↑ Phagocytosis↑	TLR-2/MAPK/NF-κB ↑	[46]
Rhein	In vitro	LPS	RAW264.7	60–140 μM	iNOS and TNFα↓	NF-κB↓; PPARγ↑	[30]
Rhein micelles	In vitro	LPS	RAW264.7	40 μM	iNOS, TNFα, IL-1β, and IL-6↓ COX-2, PGE2, NO↓	NF-κB↓	[32]
Rhein	In vitro	LPS	RAW264.7	0–35 μM	NO, IL-6↓ IL-1, HMGB1↑	NF-κB↓ IKKβ↓, Caspase 1↑	[44]
Rhein	In vitro	LPS + ATP	RAW264.7	1–20 μM	TNFα, IL-1β, and IL-6↓ COX-2↓	NF-κB↓ NALP3 inflammasome↓	[39]
Rhein	In vitro	ATP	peritoneal macrophages	0–10 μM	cytosolic Ca^2+^↓; Phagocytosis↓ ROS, IL-1β↓	P2X7↓	[42]
Chrysophanol	In vitro	LPS	RAW264.7	15 μM	iNOS, TNFα and IL-1β↓	NF-κB↓; PPARγ↑	[31]
Physcion	In vitro	_	Primary dendritic cells	1–100 μM	DCs maturation↑ Th1 differentiation↑	CD40, CD80, CD86, and MHC II↑; IL-12p70↑	[47]

**Table 2 molecules-27-03831-t002:** Effects and mechanism of AQs on digestive diseases.

Compound	Disease/Injury	Stimuli	Cell/Animal	Doses	Effects	Mechanism	Ref
Rhein	Intestinal barrier injury	LPS ip.	SD rats	ig. 66.7 mg/kg/day	intestinal damage↓ TNFα, IL-1 and IL-6↓ GSH-Px, HO-1↑	MAPK↓ Nrf2↑	[58]
Rhein	In vitro	TNFα	IEC-6	0–4 μM	TNFα, IL-1 and IL-6↓; ZO-1↑	MLCK and NF-κB↓	[59]
Rhein	Ulcerative colitis	DSS	C57BL/6J mice	ig. 50, 100 mg/kg/day, ig.	Histological changes↓ Th1, Th17↓ uric acid levels↓	Probiotic *Lactobacillus*↑	[63]
Rhein	Acute enteritis	Radiation	SD rats	ig. 90 μg/kg	NO, TNFα IL-1β and IL-6↓ MDA↓, SOD and GSH↑ Cleaved caspase-3, PARP↓	NF-κB ↓; PPARγ↑	[33]
Emodin	Colitis-associated tumorigenesis	AOM + DSS	BALB/c mice	ig. 50 mg/kg/day	Week 3: adenoma↓ Inflammatory cells infiltration↓ TNFα, IL-1α/β and IL6↓; Week 14: Dysplastic lesions↓	_	[64]
Emodin	Sepsis-induced jejunum injury	Cecal ligation and puncture	Wistar rats	ip. 10 mg/kg/day	Intestinal mucosal damage↓ TNFα, IL-6 and PCT↓ Apoptosis↓	JAK1/STAT3↑	[65]
Emodin	In vitro	Cerulein/LPS	AR42J	10–40 μM	Mitochondrial damage↓; ROS↓; TNFα, IL-6↓	ASK1/TRAF2 (JNK)↓ p38 MAPK↓	[75]
Emodin	SAP	Taurocholate	SD rats	ig. 60 mg/kg	TNFα, IL-6↓ neutrophils derived ROS↓	VDAC1↓ NLRP3 inflammasome↓	[76]
Emodin	SAP	Taurocholate	SD rats	ig. 30, 60 mg/kg	Pathological changes Amylase, Lipase TNFα, IL-6, MPO↓	P2X7↓ NLRP3 inflammasome↓	[77]
Emodin	SAP	Taurocholate	SD rats	ig. 50 mg/kg	Intestine mucosal barrier↑ Occludin, ZO-1, E-cadherin↓ Intestinal cell apoptosis ↓	Notch1, RhoA/ROCK↓ miR-218a-5p↓	[79]
Rhein	In vitro	Cerulein	AR42J	1 μM	Mitochondrial swelling and spinal fracture↓	PI3K/AKT/mTOR↓	[74]
Total Rhubarb anthraquinones	SAP	Taurocholate	SD rats	ig. 36, 72 mg/kg	Endotoxin, TNFα, IL-1β↓ Intestinal mucosal barrier↑ NO, MPO↓; Tregs, Th1/Th2↓	NLRP3 inflammasome↓	[78]

**Table 3 molecules-27-03831-t003:** Effects and mechanism of AQs on respiratory diseases.

Compound	Disease/Injury	Stimuli	Cell/Animal	Doses	Effects	Mechanism	Ref
Emodin	ALI	LPS	Wistar rats	ig. 20, 40 mg/kg	Pathological changes↓; Infiltrated inflammatory cells↓; TNFα, IL-1β, and IL-6↓	mTOR/HIF-1α/VEGF↓ NF-κB↓	[85]
Emodin	ARDS	LPS	C57BL/6J mice	ip. 5, 10, 20 mg/kg	Lung injury, inflammatory infiltration↓ Pulmonary TF, PAI-1, Collagen I, III↓ Pulmonary IL-8, IL- 1β, TNFα↓	NF-κB↓	[86]
Emodin	Lung function decreases	DEP air pollution	BALB/C mice	ip. 4 mg/kg	Lung function↑ Lipid peroxidation, ROS, GSH↓	_	[87]
Emodin	Asthma	Ovalbumin	BALB/c mice	ip. 15, 30, 60 mg/kg	Airway resistance↓ Pulmonary tissues injury↓ IL-5 and IL-17↓; IFNγ↑	Notch1-3↓	[94]
Emodin	Asthma	Ovalbumin	C57BL/6J mice	ip. 10, 20 mg/kg	Macrophages and eosinophils↓ IL-4, 5, 13, 17, NO and IFNγ↓	_	[95]
Emodin	pulmonary fibrosis	Bleomycin	SD rats	ig. 20 mg/kg	Lung structural damage↓; Collagen deposition↓ Inflammatory cell infiltration↓ Pro-inflammatory cytokines↓	NF-κB↓ Nrf2↑	[100]
Emodin	In vitro	TGFβ1	A549	60 μM	EMT↓		
Emodin	Silicosis	Silica	C57BL/6J mice	ig. 25, 50 mg/kg	alveolitis and fibrosis↓ Smad3↓	NF-κB↓ TGF-β1/Smad3↓	[101]
Chrysophanol	ALI	LPS	BALB/c mice	ip. 7.5, 15, and 30 mg/kg)	Lung injury↓ TNFα, IL-1β, IL-16, HMGB-1↓	HMGB1/NF-κB↓	[88]
Chrysophanol	Asthma	Ovalbumin	BALB/c mice	ip. 0.1, 1, 10 mg/kg	TNFα,IL-4, IL-5, IL-13↓ Airway remodeling↓	NF-κB↓	[96]
Chrysophanol	In vitro	TNFα	BEAS-2B	2, 20 μM	Proliferation↓, p-IκB↓		

**Table 4 molecules-27-03831-t004:** Effects and mechanism of AQs on arthritis.

Compound	Disease/Injury	Stimuli	Cell/Animal	Doses	Effects	Mechanism	Ref
Rhein	In vitro	PMA/urate	THP-1	1–10 μg/mL	IL-1β, TNFα, Caspase1↓	NLRP3 inflammasome↓	[106]
Rhein	In vitro	ATP	Synoviocytes of CIA rats	0.1–10 μM	Ca2+↓; ROS↓ MMP-9, COX-2 and IL-6↓	P2 × 4R	[107]
Emodin	RA	Collagen	DBA/1 mice	ip. 10 mg/kg	Synovial inflammation↓ Joint destruction↓, MMP-1, -3↓	NF-κB↓	[110]
Emodin	In vitro	LPS	Synoviocytes of RA patients	0.1–10 μM	TNFα, IL-6 and IL-8↓ COX-2, VEGF, HIF-1a↓ MMP-1, MMP-13↓	HDAC1↓	[111]
Emodin	In vitro	_	Fibroblasts of AS patients	10 µM	Caspase-3, -9↑ Atg12, Atg5, and Beclin 1↑	autophagy↑	[112]
Aloe-emodin	RA	Complete Freund’s Adjuvant	Wistar rats	50 and 75 mg/kg	paw edema volume↓ Arthritis score↓ WBC count ↓ LPO, NO↓; GSH, CAT, SOD↑	—	[113]

**Table 5 molecules-27-03831-t005:** Effects and mechanism of AQs on metabolic disorder.

Compound	Disease/Injury	Stimuli	Cell/Animal	Doses	Effects	Mechanism	Ref
Emodin	NAFLD	HFD + LPS	LDLR−/− mice	ip. 40 mg/kg	TNFα, IL-1β, IL-6, IFNγ, G-CSF, GM-CSF, MCP-1, RANTES↓; Liver leukocyte infiltration↓; Liver function↑	Erk1/2 and p38↓	[119]
Emodin	Obesity	HFD	C57BL/6 mice	ig. 80 mg/kg	glucose and insulin↓ brown AT (BAT) mass↓ systematic inflammation↓	M2 macrophage↑ TREM2↑	[120]
Emodin	T2D	HFD + streptozotocin	T2D		Hyperglycemia, dyslipidemia↓	miR-20b/SMAD 7	[121]
Emodin	In vitro	Palmitic acid	L6 myoblasts	5–20 μM	Glucose consumption↑		
Emodin	In vitro	_	C2C12/3T3-L1	6.25–50 μM	Glucose uptake, consumption↑ Glycolysis↑; lipolysis↓	NF-κB↓	[122]
Chrysophanol	Obesity	HFD	SD rats	ip. 10 mg/kg	Body weight, blood glucose↓ TG↓, HDL-C↑ IL-6, IL-1β↓; IL-10↑	AMPK/Sirt1↑	[123]
Aloe-emodin	Obesity	HFD	Wistar rats	ig. 100 mg/kg	TNFα, IL-1β, IL-6↓ VCAM1, ICAM-1↓	TLR4/NF-κB↓	[124]

**Table 6 molecules-27-03831-t006:** Effects and mechanism of AQs on cerebral vascular diseases and central nervous system.

Compound	Disease/Injury	Stimuli	Cell/Animal	Doses	Effects	Mechanism	Ref
Chrysophanol	Stroke	MCAO	CD1 mice	ip. 0.1,1, 10 mg/kg	Neurons Caspase-1, IL-1β↓ Neurological deficit↓, Brain Edema↓	NALP3↓	[131]
Chrysophanol	Stroke	MCAO	C57BL mice	ip. 0.1,1, 10 mg/kg	Survival rates↑, apoptosis↓ Neurological function↑ Pro-inflammatory cytokines↓	NF-κB↓	[132]
Aloe-emodin	Stroke	MCAO	SD rats	ig. 25, 50, 100 mg/kg	Neurological disorder↓ Infarct size↓ TNFα, MDA↓; SOD↑	PI3K/AKT/mTOR↓ NF-κB↓	[134]
Aloe-emodin	in vitro	OGD/R LPS	SH-SY5Y BV2	0–10 µM			
Emodin	in vitro	H_2_O_2_	SH-SY5Y	10–100 µM	Viability↑, Apoptosis, LDH↑	PI3K/mTOR/GSK3β	[135]
Emodin	SAE	Cecal ligation and puncture (CLP)	BALB/C mice	i.p. 20 mg/kg	Neurons apoptosis↓ Cognitive dysfunction ↓ Pathological injury↓	BDNF/TrkB↑ Autophagy↑	[137]
Emodin	sepsis-realted brain injury	LPS	BALB/c mice	ip. 20 mg/kg	Serum S100β, IL-6, TNFα, NSE↓ Brain AchE, LA↓	_	[138]
Rhein	Multiple sclerosis	EAE	Mice	ip. 5, 10, 20 mg/kg	Brain IL-2↓, Foxp3↑	Treg differentiation	[141]

## Data Availability

Not applicable.

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
