# Peer review of "Effects of Anthraquinones on Immune Responses and Inflammatory Diseases"

_molecules, 2022, doi:10.3390/molecules27123831_

Round 1

Reviewer 1 Report

The review by Xin et al concerns with the effects of anthraquinones on several diseases. While the paper regards an interesting topic, several issues should be addressed before publishing.

In particular, in several sections, mainly in paragraph 2.1 and 2.3  there is an excessive focus on biology, so some sentences should be shortened, or, alternatively, the whole review could be rewritten with more enphasis on the activity of anthraquinones.

Other minor points are:

- The language needs some polishing: i.e. in the abstract line 19 is not poperly constructed,

- The species' name should be in italic (i.e. lines 33-34),

Author Response

  1. In particular, in several sections, mainly in paragraph 2.1 and 2.3 there is an excessive focus on biology, so some sentences should be shortened, or, alternatively, the whole review could be rewritten with more emphasis on the activity of anthraquinones.

Answer: We are truly grateful to have your critical comments and thoughtful suggestions on our manuscript. According to your comments and suggestions, we have made careful modifications on the original manuscript. The 2.1 and 2.3 were simplified to emphasize the activity of AQs, and shown in the revised version.

  1. Other minor points are:

- The language needs some polishing: i.e. in the abstract line 19 is not properly constructed,

- The species' name should be in italic (i.e. lines 33-34),

Answer: The language of this manuscript has been carefully improved of the whole manuscript and marked in red. For example, the abstract of line 19 was reconstructed.

Reviewer 2 Report

The topic of the manuscript is interesting and fits well the scope of the journal. The reviewer feels it can be accepted after some minor amendments.

1) Please briefly discuss the safety profiles of Anthraquinones

2) Please briefly discuss the drug-likeness (pharmacokinetics) of Anthraquinones

3) Besides naturally occurring of Anthraquinones,  any synthetic of Anthraquinones displays better activities / safety / drug-likeness?

Author Response

1) Please briefly discuss the safety profiles of Anthraquinones

Answer: We are truly grateful to have your critical comments and thoughtful suggestions on our manuscript. According to your comments and suggestions, we have made careful modifications on the original manuscript and marked in red. We have added a paragraph to briefly demonstrate the safety and toxicity of AQs in the “Discussion” part, including the hepatotoxicity, nephrotoxicity, genotoxicity, reproductive toxicity and phototoxicity.

2) Please briefly discuss the drug-likeness (pharmacokinetics) of Anthraquinones

Answer: We have added a paragraph to briefly demonstrate the drug-likeness and pharmacokinetics of AQs referred to the reported the research and TCM database in the “Discussion” part.

3) Besides naturally occurring of Anthraquinones,  any synthetic of Anthraquinones displays better activities / safety / drug-likeness?

Answer: In addition to the exploration of natural AQs, the (semi-) synthetic derivative of AQs and the activity evaluation has always been in progress. The above-mentioned drug, Diacerein, is a semisynthetic derivative of rhein in which the two hydroxyl groups are acetylated. Other (semi-) synthetic derivative of AQs that are used in the clinical practice are mainly anthracyclines and related anticancer drugs, and Malik a et al. summarized the details of these drug (Malik EM, et al. 2016). Some newly synthetic AQs showed various effects. For example, Gecibesler et al. synthesized a new semi-synthetic AQ derivative with the NαFmoc-l-Lys and ethynyl group were synthesized based on emodin and aloe-emodin to increase the bioactivities (Gecibesler et al, 2021). The new semi-synthetic AQ shows high inhibition against HT-29 and HeLa cell lines. Further, modification of the aloe-emodin with both the ethynyl and the NαFmoc-l-Lys showed an antioxidant activity-enhancing effect.

Round 2

Reviewer 1 Report

The authors improved the manuscript, and it can be published in the present form